# Bone disease imaging through the near-infrared-II window

Chao Mi [1,2,3,4,8] ✉, Xun Zhang [1,8], Chengyu Yang[5], Jianqun Wu[5], Xinxin Chen[6], Chenguang Ma[1], Sitong Wu[1,2], Zhichao Yang[1,2], Pengzhen Qiao[1], Yang Liu[5], Weijie Wu[1], Zhiyong Guo[1,3,7], Jiayan Liao [2], Jiajia Zhou [2], Ming Guan[1,4], Chao Liang [6] ✉, Chao Liu [5,7] ✉ & Dayong Jin [1,2] ✉

Skeletal disorders are commonly diagnosed by X-ray imaging, but the radiation limits its use. Optical imaging through the near-infrared-II window (NIR-II, 1000–1700 nm) can penetrate deep tissues without radiation risk, but the targeting of contrast agent is non-specific. Here, we report that lanthanide-doped nanocrystals can passively target the bone marrow, which can be effective for over two months. We therefore develop the high-resolution NIR-II imaging method for bone disease diagnosis, including the 3D bone imaging instrumentation to show the intravital bone morphology. We demonstrate the monitoring of 1 mm bone defects with spatial resolution comparable to the X-ray imaging result. Moreover, NIR-II imaging can reveal the early onset inflammation as the synovitis in the early stage of rheumatoid arthritis, comparable to micro computed tomography (μCT) in diagnosis of osteoarthritis, including the symptoms of osteophyte and hyperostosis in the knee joint.

Mammalian bone, together with tendons and muscles, perform the vital functions of supporting the body weight and locomotion. A microscopic view of the bone structure reveals its extensive blood vessel networks and the niche for both the mesenchymal and hematopoietic stem cells, as well as their progenies[1]. Bone health is critical in maintaining body movement capability, immunity, and metabolism[2]. However, a series of commonly occurring bone diseases, including fracture, skeletal deformity, osteoporosis, rheumatoid arthritis, osteoarthritis, and bone tumor, are hard to become noticeable until the late-stage symptoms appear, unless X-ray imaging can be regularly used[3–6]. Nonetheless, regular exposure to X-radiation will cause DNA damage and leukocyte death and is classified as a "known human carcinogen" by the World Health Organization. More specifically, a study in Australia shows that exposure to an X-ray computed tomography (CT) scan (average effective radiation dose 4.5 mSv) in

childhood or adolescence leads to a 24% higher overall cancer incidence, and the incidence rate ratio increases with each additional CT scan[7]. According to the US Food and Drug Administration, 10 mSv, the effective dose of radiation from a CT scan of the abdomen and pelvis, might increase the risk of cancer by about 1 in 2000[8]. To date, there is no other alternative imaging approach for visualizing the microscopic structure and monitoring the dysfunctions of bone health in vivo.

Optical imaging allows regular and real-time visualization of cells and their functions[9–11]. More significantly, compared with the visible (400–700 nm) and near-infrared (NIR)-I (700–900 nm) ranges, light excitation and emission at the NIR-II window (1000–1700 nm) can penetrate deep tissues, which allows high spatial and temporal resolutions to be achieved with high signal-to-background ratio, as the long wavelength of light leads to minimal tissue scatterings and autofluorescence[12–15]. Towards the realization of the above potentials,

[1]UTS-SUSTech Joint Research Centre for Biomedical Materials and Devices, Department of Biomedical Engineering, Southern University of Science and Technology, Shenzhen, China. [2]Institute for Biomedical Materials and Devices (IBMD), Faculty of Science, University of Technology Sydney, Sydney, NSW, Australia. [3]National Institute of Extremely-Weak Magnetic Field Infrastructure, Hangzhou, China. [4]Shenzhen Light Life Technology Co., Ltd., Shenzhen, China. [5]Department of Biomedical Engineering, Southern University of Science and Technology, Shenzhen, China. [6]Department of Systems Biology, School of Life Sciences, Southern University of Science and Technology, Shenzhen, China. [7]Guangdong Provincial Key Laboratory of Advanced Biomaterials, Southern University of Science and Technology, Shenzhen, China. [8]These authors contributed equally: Chao Mi, Xun Zhang. ✉e-mail: mic@sustech.edu.cn; liangc@sustech.edu.cn; liuc33@sustech.edu.cn; dayong.jin@uts.edu.au

NIR-II emitting materials, including semiconductor quantum dots[16–19], carbon nanotubes[20,21], lanthanide-doped nanocrystals (LnNCs)[22–26], and organic molecules[27–31], have been developed for theranostic applications[32–34]. However, the efficient delivery of these nanoparticles to target the cellular structure of the bone remains challenging as the nanoparticles are typically filtered by the mononuclear phagocyte system, especially in the liver and spleen[35,36], not to mention the slow blood flow and the high density of bone composites[37]. These cascade barriers made the use of positive targeting ligands end up with a marginal improvement to approximately 0.9% of injected dose[38,39].

Here, we report a series of skeleton diseases in the mouse models, including sub-millimeter bone defects, rheumatoid arthritis (RA), synovitis, osteoarthritis (OA), osteophyte, and hyperostosis, which could be accurately diagnosed by the -1550 nm optical imaging, taking advantage of the spontaneous cellular transport of NIR-II agents to bone marrows.

## Results and discussions

### High-resolution NIR-II living bone imaging

We first synthesized the NaYbF$_4$: Er$^{3+}$, Ce$^{3+}$ @NaYF$_4$ nanocrystals (ErNCs) with long wavelength emission around 1550 nm, the surface coating of polyethylene glycol (PEG) can prolong blood circulation and reduce hepatic uptake (Fig. S1). Before applying ErNCs to in vivo bone imaging, we conducted a comparative experiment to prove the advantages of the 1550 nm NIR-II imaging (Fig. S2), including high resolution and tissue penetration.

After the tail-vein injection of ErNCs in mouse, the blood vessels immediately light up under NIR-II imaging, as shown in Fig. S3. Then the skeletons start to emerge while the signal from blood vessels gradually vanished in 6 h (Fig. S4). One day after the tail-vein injection, the dorsal, chest, and limb bones, including the skull, spine, sternum, tibia, phalanx, rib, and femur, have been clearly resolved with high resolution and contrast, as the in vivo images shown in Fig. 1a.

More detailed structures, including multiple segments in the spine and sternum, the growth plate, and the subchondral bone (the zoom-in area at the knee joint, confirmed by μCT in Fig. S5), can be clearly visualized. In contrast, the NIR-II imaging by NdNCs with short wavelength emissions (1064 and 1340 nm) was ambiguous (Fig. 1b and S6). Remarkably, the NIR-II signal from bones continued to increase up to 5 days after the injection (Fig. 1c, d), and long-term in vivo imaging of tibias has been achieved for more than 30 days (Fig. S7). Moreover, 3D imaging could provide more realistic and abundant morphological characteristics than two-dimensional patterns. To develop the 3D bone imaging, we utilized a multi-angle rotation method to construct the multi-view images of the mouse tibia. As shown in Fig. 1e, Supplementary Movies 1 and 2, the reconstructed stereo image vividly shows the 3D structure of the tibia and the knee joint. Compared with X-ray and CT imaging, the NIR-II imaging approach is radiation-free, and allows for the long-term, non-invasive, and time-critical diagnosis of skeletal disorders associated with bone diseases.

### Passive targeting of nanoparticles to bone marrow

We find that nanoparticles' passive transportation by marrow macrophages and endothelial cells underpin the remarkable bone-targeting ability. Such spontaneous cellular transport of nanoparticles has profound significance on biomedical applications, including bone imaging and drug delivery[5]. Recent research argues that the passive targeting is mainly due to the unique hydroxyapatite mineral binding ability of PEG-coated nanoparticles[40]. In our study, the fluorescence from bone vanished after flushing out the bone marrow, while marrow mesenchyme showed a strong signal (Fig. 2a). Furthermore, after decalcification, the NIR-II signal was dramatically improved compared to the undecalcified bones (Fig. 2b). These results confirmed that ErNCs were mainly accumulated in bone marrow, rather than bonding with the cortical bone.

This motivated us to investigate deeply into the bone marrow targeting mechanism. By co-localization 3D confocal microscopic imaging on stained marrow sections, we confirmed that the Cy3 fluorescence of ErNCs@Cy3 overlaps with the channel of macrophages (Fig. 2c and S8) with a Pearson correlation coefficient of up to 0.57, which suggests the macrophage's uptake of ErNCs. This is because plasma proteins (e.g., immunoglobulins, adhesion mediators, complement proteins) may act as opsonin to bind with ErNCs, leading to efficient clearance by the mononuclear phagocyte system in the liver, spleen, bone marrows, and other organs[36,41,42]. Considering the slow blood flow in the dense capillary network inside marrows[37,43], the local macrophages take their time to capture the escaped nanoparticles from the liver and spleen, and then transport them to a distance in the marrows[44]. Studies suggested that up to 97% of nanoparticles may enter tumors through endothelial cells in blood vessels[45]. Here, we proved a similar process in marrows, as confirmed by the fluorescence co-localization results in Fig. 2d. In addition, the whole area of the cortical bone shows no Cy3 signal at all (Fig. 2e).

### Toxicity studies and intact nanocrystals' body clearance

We carefully examined the potential toxicity that may be caused by the accumulation of ErNCs in bone marrow. As shown in Fig. S9, the H&E staining tissue sections show no apparent damage or abnormality on cellular structures after 14 and 70 days post-injection of ErNCs. Moreover, we monitored the stable NIR-II fluorescence from feces for 13 days after the injection, suggesting that the nanoparticles were gradually excreted (Fig. S10). The cellular toxicity study further proves the prepared ErNCs are biocompatible and safe for in vivo NIR-II imaging (Fig. S11).

### In vivo NIR-II detection of bone defects for up to 11 days

To evaluate the imaging resolution, we purposely operated the -1 mm mono-cortical bone defects on the mice tibias (Fig. 3a)[3,46]. The X-ray imaging proves the injured bone sites with around 1 mm diameter (Fig. 3b). Impressively, the NIR-II imaging diagnoses also showed a similar result of 1.2 mm (Fig. 3a, b). Notably, the fluorescent intensity drop within 6 mm around the injured site could be caused by impaired blood circulation.

As the surgery causes collateral damage to the soft tissue and induces poor blood circulation of ErNCs on the surgical site, we postponed the injection of ErNCs by 5 days after the bone defect surgery to allow the damaged vascular network to heal. As shown in Fig. 3c, long-term monitoring from Day 6 to 11 after the bone defect surgery showed a shaded area that matched the size of the bone defect. The control group with the sham surgery showed slight abnormality around the soft tissue wound on Day 6 after surgery. The cross-sectional intensity further confirms the size of the bone defect on the left hindlimb as 1.2 mm (Fig. 3d), compared with the result of no defect being detected on the right hindlimb 11 days after surgery, as the bone injury recovery is usually slower than soft tissue wound healing (Fig. S12). More cases of NIR-II imaging, shown in Fig. S13, accurately resolved the sizes of the bone defects as 1.2, 1.36, and 1 mm, arbitrarily made in surgery.

### High-specificity inflammation imaging to reveal the early onset and progression of rheumatoid arthritis (RA)

RA is a chronic autoimmune skeleton disease with a worldwide annual incidence of 3 cases per 10,000 and a prevalence rate of 1%[47,48]. NIR-II imaging on collagen-induced arthritis (CIA), the gold standard animal model for RA, illustrated that two toes displayed much higher NIR-II intensity compared with the other toes from one mouse paw with mild RA. With severe RA, the heavily swollen paw was overexposed in all the toe joints under NIR-II imaging, in sharp contrast to the clearly visualized bones and joints of the healthy mouse paw (Fig. 4a and S14). The μCT analysis consistently showed the aggravation of RA (Fig. 4b–d), as evidenced by 3D images of paws and bone mineral density (BMD) and a

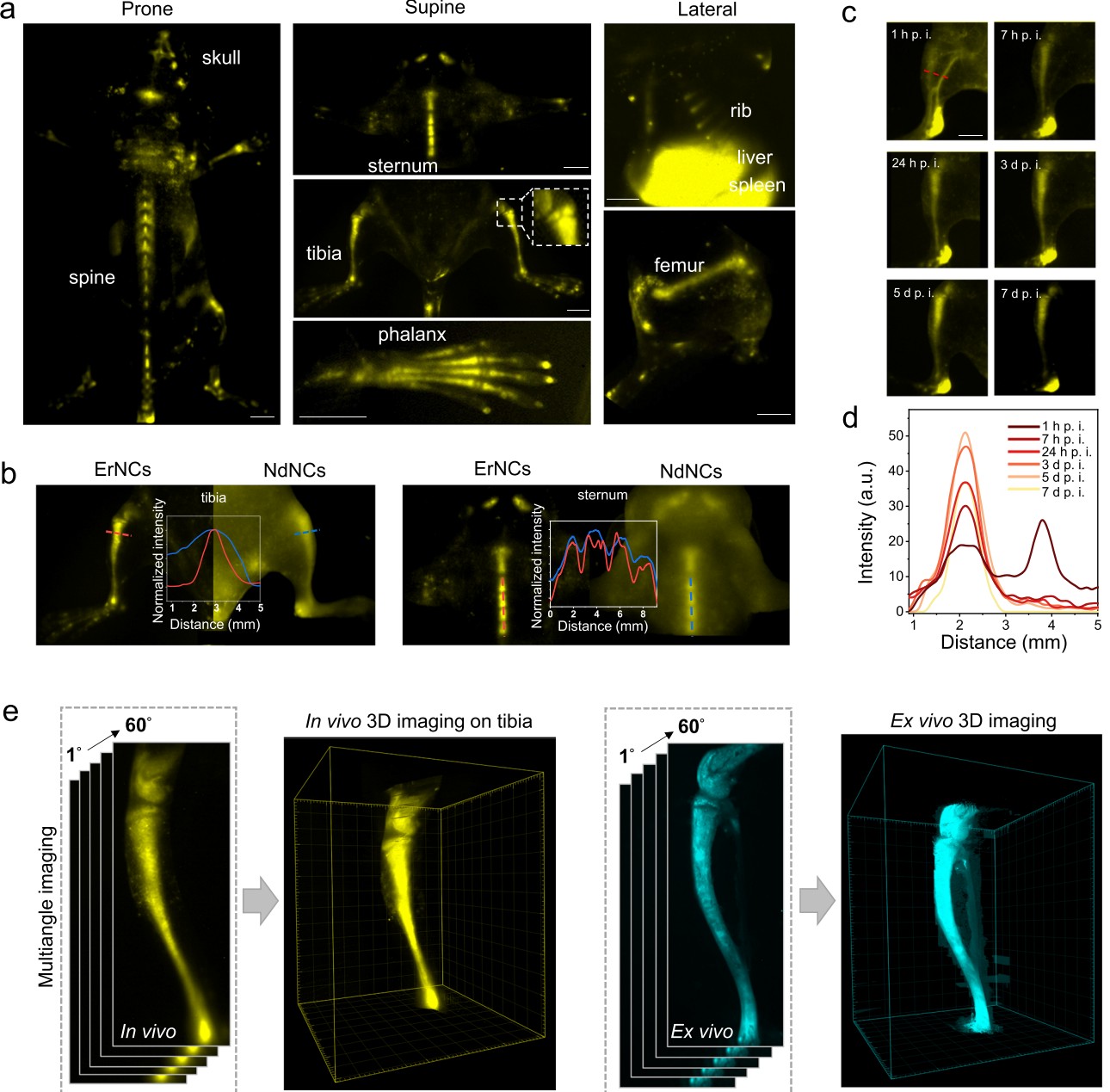

**Fig. 1 | High-resolution NIR-II imaging of mice bones in both two and three dimensions. a** NIR-II in vivo imaging of murine bones in prone (skull, spine), supine (sternum, tibia, phalanx, subchondral bone in the zoom-in picture), and lateral (rib, femur) postures by the 1550 nm fluorescent ErNCs (980 nm excitation, 38 mWcm$^{-2}$, 1319 nm long-pass). Scale bars: 2 mm. **b** The NIR-II imaging of bone and corresponding cross-sectional intensity profiles by ErNCs (980 nm excitation, 38 mWcm$^{-2}$, 1319 nm long-pass) and NdNCs (808 nm excitation, 34 mWcm$^{-2}$, 900 nm long-pass), respectively. **c** In vivo NIR-II imaging of mouse tibia by ErNCs (980 nm excitation, 38 mWcm$^{-2}$, 1319 nm long-pass, 60 ms) at the different post-injection time points (d: day). Scale bar: 2 mm. **d** Cross-sectional intensity profiles along the same position of the time series images represented by the red dash line in **c**. **e** The 3D NIR-II imaging on mouse tibia reconstructed by a series of images recorded under different angles (0°–60°) rotated along the central axis of the tibia.

microarchitecture parameter of bone (BV/TV), which proves the potential of NIR-II imaging for monitoring RA progression.

**Accurate recognition of synovitis**

Early diagnosis and treatment of RA are limited by its unknown etiology and initial similarity to other inflammatory diseases[49]. The pathophysiology of RA starts with chronic inflammation of the synovial membrane, then erosion of articular cartilage and juxta-articular bone[48]. Thus, early diagnosis of RA depends on the accurate recognition of synovitis.

Significantly, the NIR-II imaging has high specificity and sensitivity to distinguish inflammation. As shown in Fig. 4e and S15a, in one same mouse toe, the inflamed toe joint always exhibits much higher intensities than the adjacent normal joint within 3 mm, which is confirmed by immunofluorescence imaging of inflammatory factor IL-1β (Fig. 4f and S15b). Furthermore, the sharp NIR-II images clearly show the position of the strong signal overlaps with the distribution of synovium on the side of the joint (Fig. 4g and S15c), which excludes similar inflammatory diseases and is determined as synovitis.

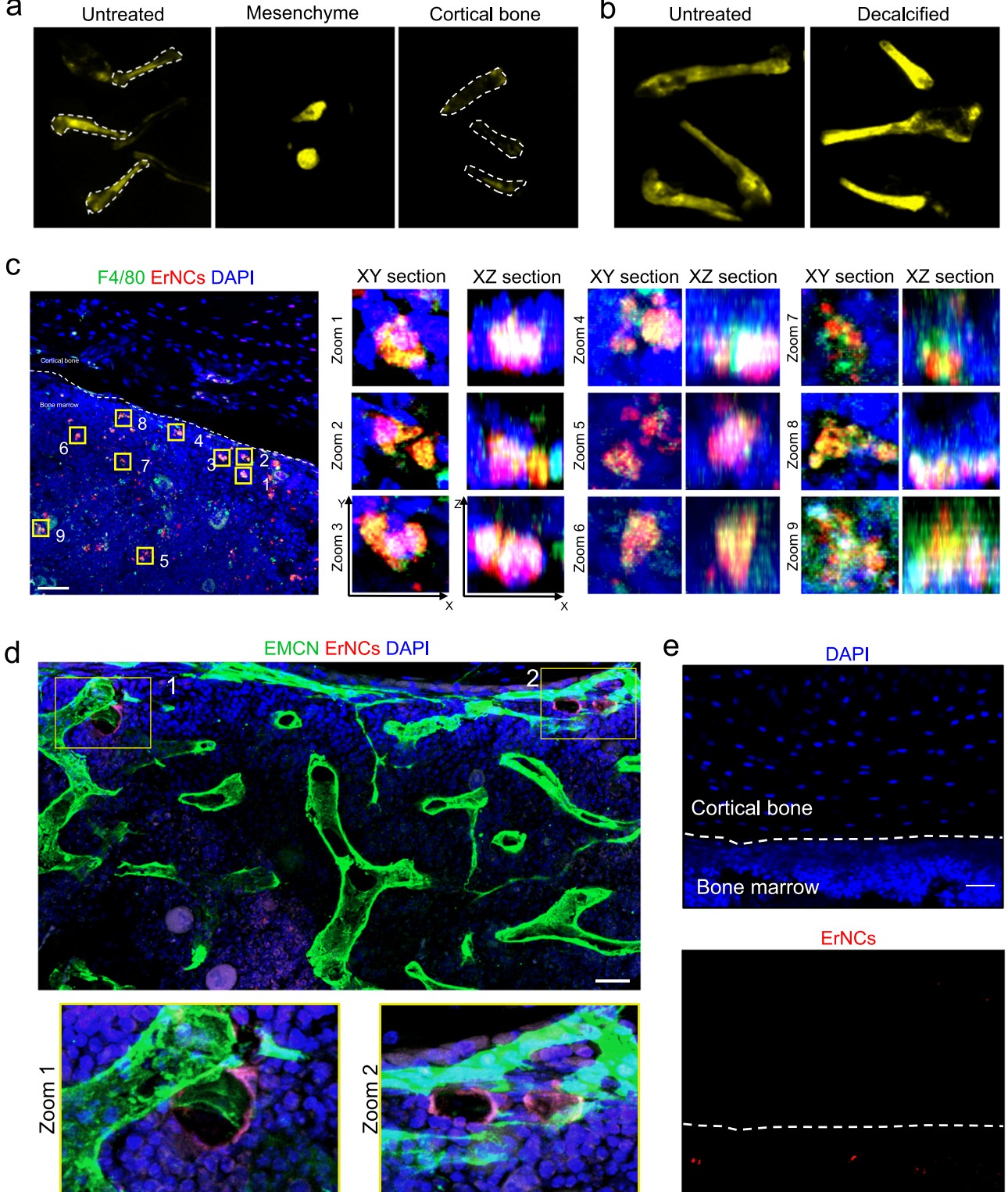

**Fig. 2 | Cell uptakes of ErNCs in mouse bone marrow. a** Ex vivo NIR-II imaging of the untreated mice femurs, the marrow mesenchyme of the femurs, and the femurs without mesenchyme after administration of ErNCs (980 nm excitation, 38 mWcm$^{-2}$, 1319 nm long-pass, 200 ms). **b** Ex vivo NIR-II imaging comparison between the undecalcified bones and the decalcified bones after administration of ErNCs (980 nm excitation, 38 mWcm$^{-2}$, 1319 nm long-pass, 300 ms). **c** 3D confocal microscopic imaging on the stained tibia sections collected from a mouse 36 h after ErNCs@Cy3 injection, including nine zoom-in corresponding XY and XZ sections of interest. Green channel: F4/80 labeled macrophages. Red channel: ErNCs@Cy3.

Blue channel: DAPI labeled cell nucleus. The experiment was repeated three times independently, with similar results. Scale bar: 80 µm. **d** Confocal images of bone marrow in the stained tibia sections collected from a mouse 1 h post ErNCs@Cy3 injection, including two zoom-in regions of interest. Green channel: Endomucin (EMCN) labeled endothelial cells. Red channel: ErNCs@Cy3. Blue channel: DAPI labeled cell nucleus. The experiment was repeated three times independently, with similar results. Scale bar: 30 µm. **e** Confocal images of the stained cortical bone area from the same mouse tibia in **c**. The experiment was repeated three times independently, with similar results. Scale bar: 30 µm.

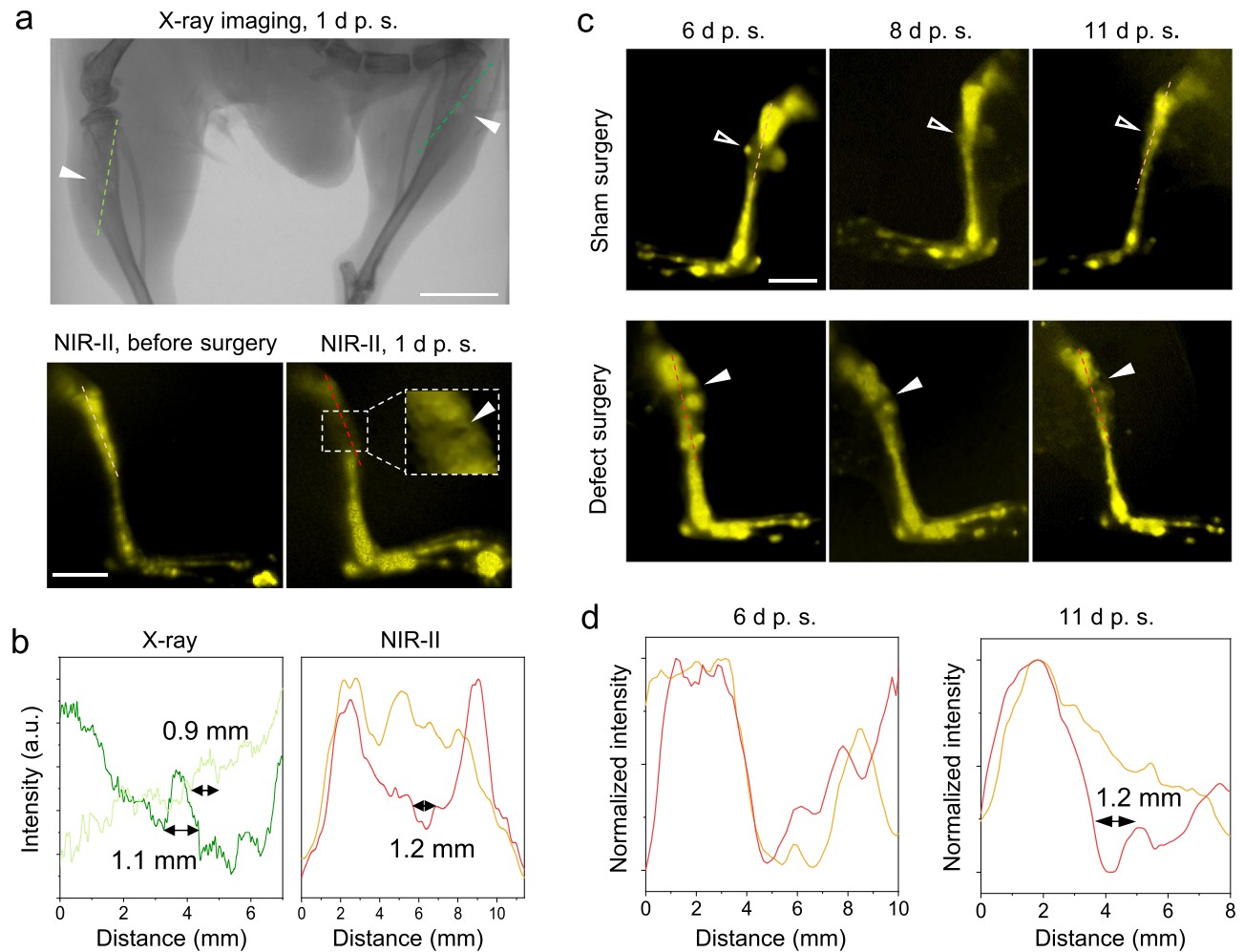

**Fig. 3 | Bone defects (~1 mm) on mice tibias diagnosed by X-ray and NIR-II imaging. a** Comparison between the X-ray and NIR-II in vivo imaging on the tibial defects (solid arrowheads). The X-ray imaging was measured 1 d post-surgery (p. s.), and the NIR-II imaging was taken before and 1 d after surgery, respectively. Scale bars: 5 mm. **b** The intensity profiles along the cross sections are indicated by the corresponding-colored dash lines in **a**. The measured sizes of the tibia defects were given. **c** The NIR-II in vivo imaging on the right (sham surgery as control) and left hindlimb (defect surgery) of a mouse on Days 6, 8, and 11 after surgery. The right hindlimb has a soft tissue wound (open arrowheads) from a sham surgery, while the left hindlimb contains both the soft tissue wound and the bone defect (solid arrowheads). Injection of ErNCs was 5 days after surgery via the tail vein. The experiment was repeated three times independently, with similar results. Scale bar: 5 mm. **d** The intensity profiles along the cross sections are represented by the dash lines in both the sham surgery group (orange) and bone defect surgery group (red) in **c**.

## NIR-II and μCT imaging diagnosis on osteophytes and hyperostosis in the osteoarthritis (OA) mouse model

OA is the most common degenerative joint disease and the leading cause of physical disability. It occurs with the formation of osteophytes, cartilage loss and so on[4,50], as illustrated in Fig. 5a. To demonstrate the power of NIR-II imaging in diagnosing OA, we established a mouse model induced by anterior cruciate ligament transection (ACLT) surgery. The μCT and safranin O-fast green staining results (Fig. 5b) confirmed the gradual cartilage degeneration and the bone mass decrease during the formation of OA. As shown in Fig. 5c, NIR-II imaging can give similar results with μCT, including not only the normal morphology of the right knee joint as control, but also the formation of the millimeter-sized osteophyte at the left knee joint after ACLT. However, the μCT takes about 1 h to capture the knee joint cross sections and extra time for reconstruction, which is much slower than NIR-II imaging with hundreds-millisecond exposure time.

Interestingly, the NIR-II optical imaging could also reveal the growth of hyperostosis. As shown in Fig. 5d, the NIR-II images show both femur and tibia in all three healthy knee joints, however, only the femur in the left knee with OA cannot be imaged as clearly. The

intensity gap Δ2 in the knee with OA is over 2 times higher than Δ1 in all three control cases (Fig. 5e). These results strongly indicate the formation of hyperostosis in the cortical bone[4,50], as the thicker cortical bone will block more NIR-II signal penetrated out the femur marrow. Consistent with this view, the μCT cross sections agree with the hyperostosis revealed by NIR-II imaging, as the medial condyles of the left femurs show increased bone thickness over 200 μm after ACLT, while the right femurs as control are generally thinner than 100 μm (Fig. 5f, g and Figs. S16–S18). Contrastively, it is highly possible that ACLT causes no hyperostosis growth at the tibia because the NIR-II intensities on the tibia are similar between the control and ACLT groups. The μCT proves this again as both groups show normal and similar bone thicknesses for tibias (Figs. S19 and S20).

While the NIR-II in vivo imaging has attracted enormous attention in the past decade, investigations on the efficiency of the NIR-II fluorescence agents, their bone-targeting mechanisms, body clearance pathway, biocompatibility, and toxicity are still in their infancy[51,52]. This study has realized non-invasive, high-flexibility, and high-resolution imaging diagnoses of small cortical bone defects, rheumatoid arthritis, and osteoarthritis. Compared with the conventional X-ray and μCT

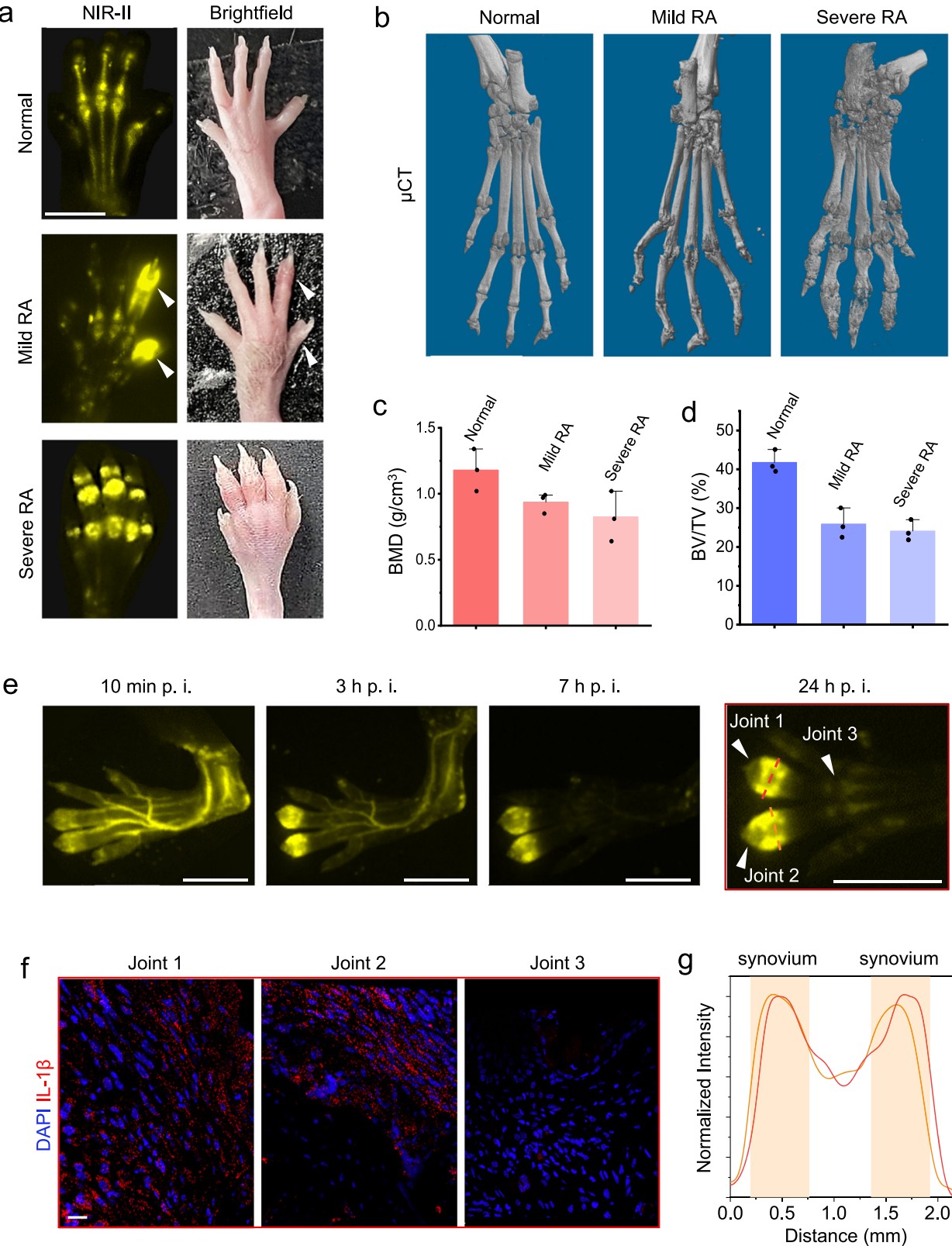

**Fig. 4 | Early recognition of rheumatoid arthritis (RA) through accurate inflammation NIR-II imaging. a** The hind paws from a normal mouse, a CIA model mouse with mild RA, and a CIA model mouse with severe RA under NIR-II imaging and bright field, respectively. The experiment was repeated three times independently, with similar results. Scale bars: 5 mm. **b** Representative μCT images of hind paws from the normal mouse, CIA mouse with early-stage arthritis, and CIA mouse with late-stage arthritis. Scale bars: 5 mm. **c**, **d** Quantitative μCT analyses of bone mineral density (BMD) (**c**) and the bone volume fraction (BV/TV) (**d**) in toe joints shown in (b), respectively. **e** Time series NIR-II imaging on the CIA mouse hind paw with early-stage arthritis. Scale bars: 5 mm. **f** Immunofluorescence analyses of inflammatory factor IL-1β in the synovium tissues from three toe joints (numbered arrowheads) in the hind paw in **e**. Scale bar: 20 μm. **g** The normalized intensity profiles along the cross sections are indicated by the dash lines in **e**.

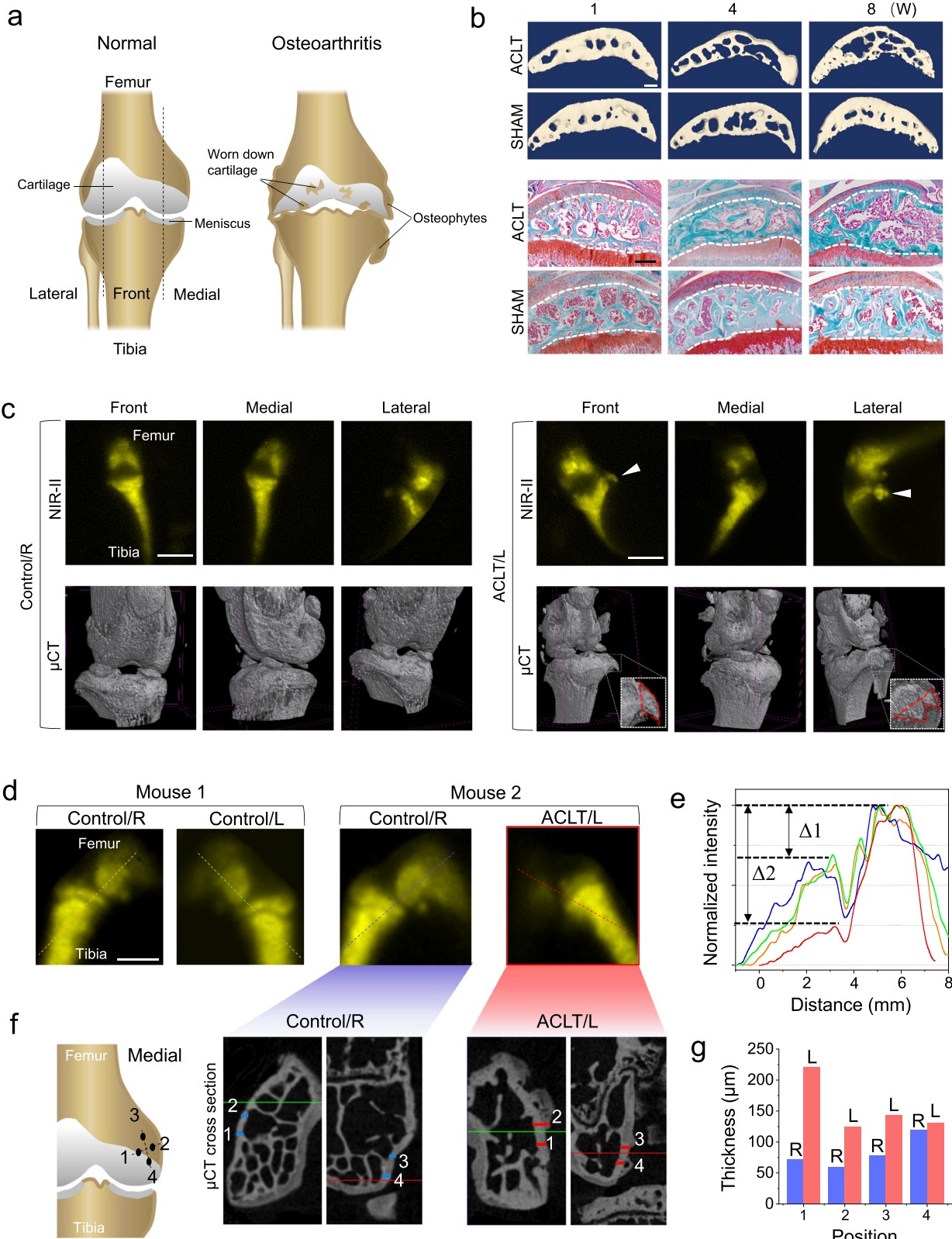

techniques, NIR-II bone imaging can not only satisfy the rapid test and frequent usage but also provide detailed incidence features and achieve matchable diagnosis results. More prospectively, it is free of radiation risk for regular use and long-term monitoring of potential bone disorders and treatment progression. The rapid progress made in high-resolution optical imaging systems and high-efficiency optical materials, as well as the ongoing efforts in studying in vivo specific targeting of nanoparticles with different sizes and surface conditions with improved body clearance, including the cell kinetics during the capture and transport of nanoparticles in the marrow, will continue to advance the field of NIR-II imaging towards pre-clinical and clinical translations.

**Fig. 5 | In vivo NIR-II imaging on symptoms of osteoarthritis compared to μCT.**
**a** The illustration of osteoarthritis v.s. normal arthrosis. **b** Representative reconstructed μCT images of coronal sections of medial tibia plateaus (first and second rows) and safranin O-fast green staining of tibia plateaus (third and fourth rows), during the time-course of 1, 4, and 8 weeks after ACLT and sham surgery, respectively. The experiment was repeated three times independently, with similar results. Scale bars: 200 μm. **c** In vivo NIR-II imaging and the corresponding ex vivo μCT images on both hind knee joints of the same mouse (in three angles of view), the left knee joint (ACLT/L) had an ACLT surgery 8 weeks ago while the right (Control/R) as the control has no surgery. The solid arrowheads and the red dash lines indicate an osteophyte in the left knee. Scale bars: 5 mm. **d** In vivo NIR-II imaging on the knee joints from two mice, only the left knee of Mouse 2 (ACLT/L) had an ACLT surgery 8 weeks ago, two knees of Mouse 1 and the right knee of Mouse 2 were untreated as control (L: left, R: right). The experiment was repeated three times independently, with similar results. Scale bar: 2 mm. **e** The normalized intensity profiles along the cross sections are indicated by the dash lines in **d**. (Orange: Right knee of Mouse 1, green: Left knee of Mouse 1, blue: Right knee of Mouse 2, red: Left knee of Mouse 2). **f** The ex vivo μCT cross-section images on the left and right femurs of Mouse 2 in (**d**). The locations of the cross sections on the femurs are illustrated as well. **g** Comparison of the cortical bone thicknesses of all positions indicated by the numbered short lines (blue: Right femur; red: Left femurs) in **f** from both femurs of Mouse 2 (L: left, R: right).

## Methods
Our research complies with all relevant ethical regulations, which are approved by the Animal Care Committee of the Laboratory Animals at Southern University of Science and Technology.

### Materials
$YCl_3 \cdot 6H_2O$ (99.99%), $YbCl_3 \cdot 6H_2O$ (99.99%), $ErCl_3 \cdot 6H_2O$ (99.99%), $CeCl_3 \cdot 6H_2O$ (99.99%), $NdCl_3 \cdot 6H_2O$ (99.99%), $NH_4F$ (99.99%), NaOH (99.9%), oleic acid (OA, 90%), 1-octadecene (ODE, 90%), poly(maleic anhydride-alt-1-octadecene) (PMH; 3000-5000 m. W.), 4-(dimethylamino)pyridine (DMAP), 4-morpholineethanesulfonic acid (MES), tris hydrochloride (Tris-HCl), 1-(3-dimethylaminopropyl)−3-ethylcarbodiimide hydrochloride (EDC) were purchased from Sigma-Aldrich. Cyclohexane, chloroform, ICG were bought from Aladdin Co. Ethanol (99.7%) and methanol (99.7%) were purchased from Shenzhen hs-science Co. Methoxy polyethylene glycol amine ($mPEG-NH_2$; 5000 m.W.) was purchased from Laysan-Bio and $Cy3-mPEG-NH_2$ (2000 m.W.) was purchased from Xi'an ruixibio Co. All reagents were used as received without further purification.

### Synthesis of lanthanide-doped nanocrystals (LnNCs)
LnNCs (Ln=Er, Nd) were synthesized by the coprecipitation method. Taking $NaYbF_4$: 4% $Er^{3+}$, 4% $Ce^{3+}$ as an example, 1 mmol $RECl_3 \cdot 6H_2O$ (RE = Yb, Er, Ce) with the molar ratio of 92:4:4 was added to a 50 mL flask containing 6 mL OA and 15 mL ODE. The mixture was heated to 160 °C under argon for 30 min (min) to obtain a clear solution and then cooled down to room temperature. Then 8 mL methanol solution of 6 mmol NaOH was added to the mixture followed by 15 min stirring. By heating the solution to 100 °C for 15 min, the methanol was removed under argon. Again, the mixture was cooled down to room temperature. After that 8 mL methanol solution of 4 mmol $NH_4F$ was added to the mixture followed by 30 min stirring. Then the temperature was set at 100 °C and the solution was heated for 30 min to remove methanol, next the solution was further heated to 300 °C for 40 min. Finally, the reaction solution was cooled down to room temperature, and $NaYbF_4$: 4% $Er^{3+}$, 4% $Ce^{3+}$ nanocrystals were precipitated by ethanol and washed with cyclohexane, ethanol, and methanol 3 times. $NaYF_4$: 3% $Nd^{3+}$ nanocrystals were prepared by a similar method.

### Epitaxial-growth to form $NaYF_4$-coated core-shell structure LnNCs
$NaYF_4$ shell precursors were prepared first. Firstly 1 mmol $YCl_3 \cdot 6H_2O$ were added to a 50 mL flask containing 6 mL OA and 15 mL ODE. The mixture was heated to 160 °C under argon for 30 min to obtain a clear solution and then cooled down to room temperature, followed by the addition of 8 mL methanol solution of $NH_4F$ (4 mmol) and 5 mL methanol solution of NaOH (2.5 mmol). Then after stirring for 30 min, the temperature was set at 100 °C and the solution was heated under argon for 30 min to remove methanol, and the solution was further heated to 150 °C for another 30 min. Finally, the reaction solution was cooled down to room temperature as $NaYF_4$ shell precursors.

The epitaxial growth of LnNCs (Ln=Er, Nd) to form a core-shell structure was realized by a hot-injection method. Taking $NaYbF_4$: 4% $Er^{3+}$, 4% $Ce^{3+}@NaYF_4$ for example, 0.2 mmol $NaYbF_4$: 4% $Er^{3+}$, 4% $Ce^{3+}$ core samples in cyclohexane were added to a 50 mL flask containing 4.5 mL OA, 11.5 mL ODE. The mixture was heated to 150 °C under argon for 20 min, and then the solution was further heated to 300 °C. After that, 5 mL of $NaYF_4$ shell precursors in total were injected into the reaction mixture step by step with an injection rate of 0.2 mL every 2 min. After the injection of 5 mL $NaYF_4$ shell precursors, the reaction solution was ripened at 300 °C for 5 min. Finally, the reaction solution was cooled down to room temperature, and $NaYbF_4$: 4% $Er^{3+}$, 4% $Ce^{3+}@NaYF_4$ nanocrystals were precipitated by ethanol and washed with cyclohexane, ethanol, and methanol. NdNCs were prepared in the same method by using $NaYF_4$: 3% $Nd^{3+}$ nanocrystals as the core samples.

### Preparation of PEG-coated LnNCs
The OA-capped LnNCs (30 mg) in 1 mL cyclohexane was added with 3 mL chloroform. Then 80 mg PMH was added to the mixture under ultrasonic conditions and kept stirring for 10 h (h). After that, the solution was evaporated by rotary evaporation for 15 min and transparent precipitation appeared at the bottom. Next, 40 mg DMAP dissolved in 4 mL water was added to dissolve the residue again under ultrasonic conditions for 30 min. Then the solution containing PMH-coated LnNCs was centrifuged at 15000 rpm for 30 min and the sediment was washed 2 times with MES solution (10 mM; pH 8.5) to remove excess DMAP and PMH.

After that, 2 mg EDC dissolved in 200 μL water was then added to the 4 mL MES sample solution and sonicated for 15 min. 4 mg $mPEG-NH_2$ dissolved in 2 mL MES solution (10 mM; pH = 8.5) was added into the above solution, and shaken for 3 h. 3 mg Tris-HCl dissolved in water was added, and then the solution was shaken for another 1 h. Finally, the solution was centrifuged at 4400 rpm for 5 min and the supernat containing PMH-PEG-coated LnNCs was washed with a centrifugal filter (100 K) 2 times by saline.

### Preparation of Cy3-coated ErNCs
The Cy3-coated ErNCs were prepared by a similar process, instead of the 4 mg $mPEG-NH_2$ were replaced by 4 mg $Cy3-mPEG-NH_2$.

### Characterization techniques
The morphology of the formed nanocrystals was characterized by transmission electron microscopy (TEM) imaging (Hitachi HT7700) with an operating voltage of 100 kV. The samples were prepared by placing a drop of a dilute suspension of nanocrystals onto copper grids.

The NIR-II luminescence spectra for all samples were obtained by using a spectrometer (OmniFluo990-MLNIR02, Beijing Zolix) equipped with a NIR-II PMT (H10330C-75-C4, HAMAMATS). A 980 nm CW diode laser (Changchun New Industries Optoelectronics Technology) was used for LnNCs (Ln=Er, Ho, Pr, Tm) measurements, and an 808 nm laser was used for NdNCs and ICG spectra measurements. The OA-capped core and core-shell ErNCs in cyclohexane, and PEG-capped core-shell ErNCs in water were prepared to a concentration of 5 mg/mL for spectra intensity comparison under a 20 $mWcm^{-2}$ excitation density.

## Cytotoxicity assay

The viability of 4T1 cells was evaluated by a standard CCK-8 assay. The cell proliferation and cytotoxicity assay kit CCK-8 was obtained from Beijing Solarbio Science & Technology Co., Ltd. The cells were seeded into 96-well plates at a density of $2 \times 10^3$ cells/well. After incubation overnight, the cells were further incubated in a fresh culture medium containing different concentrations of PEG-ErNCs (0, 0.25, 2.5, 25, 100, 250, 500, and 1000 µg/mL) for another 18 h. Then, 10 µL of CCK-8 solution was added to each well. After incubation for another 1.5 h, the absorbance at 450 nm was measured using a spectrophotometer (Varioskan LUX, Thermo Fisher Scientific, USA). The cell viability was calculated from the ratio of the absorbance value of the PEG-ErNCs treatment group to that of the untreated group.

## Animal

All mice were purchased from Guangdong Medical Laboratory Animal Center (Guangdong, China), including 6-week-old male DBA/1 mice for collagen-induced arthritis (CIA) preparation and 6-week-old male BALB/c mice for other experiments. The mice were grown in an animal facility under filtered air conditions (21–22 °C) in plastic cages with sterilized wood shavings for bedding and provided pure water. The mice were raised in a specific pathogen-free (SPF) environment. Before NIR-II mouse imaging, the mouse hair was removed carefully by depilatory creams under anesthetization. All animal experiments were strictly performed under the guidelines of the Chinese Council for Animal Care, approved by the Animal Care Committee of the Laboratory Animals at Southern University of Science and Technology.

## Bone defect surgery

Under isoflurane anesthesia, a small skin incision was made over the midline of the anteromedial aspect of the tibia to build bone defect. A 1.0 mm circular defect centered between the tibiofibular junction and the tibial tubercule was made using a precision surgical drill. The defect center was located 4.3 mm below the proximal articulating surface of the proximal tibia. After saline irrigation, the incision was closed with 7-0 nylon sutures, and mice were transferred into a clean cage positioned on a 37 °C heating pad.

## Collagen-induced arthritis (CIA) model preparation

CIA is the gold standard animal model for rheumatoid arthritis (RA). In our experiments, CIA was induced by immunization with bovine type II collagen (2 mg/mL, Chondrex Inc.) emulsified in Complete Freund's adjuvant (CFA, 4 mg/mL, Chondrex Inc.). A homogenizer with a small blade was used to emulsify the CFA with the collagen solution. Firstly, seal the tip of the syringe with a 3-way stopcock. Next, place the syringe in the ice bath to keep the emulsion cool during mixing. Add the same volume of CFA and collagen to the end of the syringe sealed with the 3-way stopcock. Then continue mixing the emulsion at a maximum speed of the homogenizer for 2 min. After that, cool down the emulsion in the ice bath for 5 min. Repeat the mixing and cooling for the other 3 times until obtain stable emulsion, which will not diffuse in water. Finally, a 6-week-old male DBA/1 mouse was immunized with 100 µL of the prepared emulsion by tail-vein injection and CIA usually develops between 21–28 days after injection.

## Anterior cruciate ligament transection (ACLT) surgery

For ACLT surgery, the mice were anesthetized first and a small skin incision was made over the side of the knee joint to allow to find the anterior cruciate ligament. The anterior cruciate ligaments were transected surgically to induce mechanical instability-associated osteoarthritis (OA) in 8 weeks. Then the incision was closed with 7-0 nylon sutures, and mice were transferred into a clean cage positioned on a 37 °C heating pad.

For sham surgery as control, only the small skin incision was made and then closed with 7-0 nylon sutures same with the ACLT surgery, with no surgery to the anterior cruciate ligament.

## NIR-II planar and 3D optical bone imaging

The mice were first anesthetized and then injected with the NIR-II contrast agent. A NIR-II animal imaging system (NIROPTICS Ltd., China) equipped with an InGaAs camera (Photonic Science, UK) and laser source was used for the NIR-II fluorescent Imaging. Emission from ErNCs under 980 nm laser excitation was typically collected with a 1319 nm long-pass filter, and emission from NdNCs under 808 nm laser excitation was collected with a 900 nm long-pass filter.

The 3D images were taken by using a homemade NIR-II whole-animal light sheet imaging system with a NIR-II camera (C-RED2, First Light Advanced Imagery). The excitation source is a 980 nm laser with tunable power up to 5 W. The emission signals from the animal were filtered through a 1319 nm long-pass filter (Chroma Technology Corporation). For in vivo imaging, the anesthetized mice were placed on a rotating platform, and for ex vivo the soft tissue on the tibia was removed. During the measurement, the tibias of the mice were fixed and well coincident with the axis of rotation. The horizontal position was set as 0° and perpendicular to the camera, then the automatic stage was rotated to both sides for 30° respectively with the step of 1°, and the three-dimensional stack of 61 NIR-II images was taken at all angles. Finally, the 3D images were obtained by spatial rotation transformation and interpolation algorithm and displayed by Imaris software.

## In vivo and ex vivo X-ray and µCT bone imaging

For in vivo X-ray and µCT imaging, the mice were anesthetized and fixed in the chamber and a small animal µCT (SkyScan 1276, Bruker) was used for the test. For the ex vivo test, the bone was dissected and the soft tissue on the bone was removed. The metaphyseal areas of the tibia and femur were scanned by a µCT scanner (Skyscan 1172, Bruker). The 3D reconstructions of the knee joint were performed by Mimics 17 (Materialize, USA).

## Bone immunofluorescence staining for nanoparticle localization and inflammation test

Fresh tibias and hind paws were fixed in 4% paraformaldehyde at 4 °C for 4 h and decalcified in 0.5 M EDTA at 4 °C for 24 h. The tibias were cryo-protected in 20% sucrose solution at 4 °C for 24 h, then embedded in a gelatin-based medium and stored at −80 °C. Samples were cryosectioned into 80 µm thick tissue slices along the coronal plane using a cryostat (Leica CM1950, Weztlar, Germany). Tibia sections were stained with endomucin (EMCN, 1:200, sc-65495, Santa Cruz Biotechnology, USA), anti-F4/80 antibody (ab6640, Abcam, UK) followed by Alexa Fluor 488 secondary antibodies from donkey (A32790, 1:300, Thermo Fisher Scientific, USA). Toe joint sections were stained with a rabbit anti-IL-1β (ab283818, Abcam, UK), and Alexa Flour 594 labeled goat anti-rabbit IgG (ab150080, Abcam, UK) was used as the secondary antibody. The slices were mounted with DAPI Fluoromout-G (0100-20, SouthernBiotech, Birmingham, USA) and coverslipped.

## Confocal imaging on immunofluorescence-stained bone tissue slices

Nikon A1R confocal microscope (Tokyo, Japan) was used to acquire 3D fluorescent images with a 20× objective lens, including optical channels of EMCN and F4/80 (500 to 530 nm filter) with 488 excitations, Cy3 (570 to 617 nm filter) with 561 nm excitation, and DAPI (425 to 475 nm filter) with 405 nm excitation, respectively. Z-stacks of 40 µm in thickness with a step of 2 µm were taken, and the $x–y$ resolution is 0.624 µm.

## Histological analysis

Mouse knee joints were fixed in 4% paraformaldehyde for 24 h, and decalcified in 10% ethylenediaminetetraacetic acid (EDTA) for 3 weeks, dehydrated in graded ethanol, and then embedded in paraffin for histological analyses.

Sagittal sections were obtained at a 6 μm thickness using a rotary microtome (Leica RM2255; Leica, Wetzlar, Germany). The paraffin sections of the organs were stained with Safranin O/Fast Green kit (G1371, Solarbio, China) or Hematoxylin and eosin (H&E) kit (G1031, Solarbio, China) according to the staining kit instructions.

## Reporting summary

Further information on research design is available in the Nature Portfolio Reporting Summary linked to this article.

## Data availability

The authors declare that other data related to this research are available within the paper and its Supplementary Information or from the authors upon request. Source data are provided in this paper.

## Code availability

Codes for in vivo and ex vivo 3D bone imaging reconstruction algorithms are provided in this paper (https://github.com/MNBC333/Codes).

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

## Acknowledgements

The authors acknowledge the financial support from the National Natural Science Foundation of China (62005179, C. Mi, 82172386, C. Liang, 81922081, C. Liang), China Postdoctoral Science Foundation (2020M682866, C. Mi), the Shenzhen Science and Technology Program (KQTD20170810110913065, D. Jin, 20200925174735005, D. Jin), Shenzhen Science and Technology Innovation Commission (KQTD20200820113012029, C. Liu, JCYJ20190809114209434, C. Liu, JCYJ20210324104201005, C. Liang), the Department of Education of Guangdong Province (2021KTSCX104, C. Liang), and Guangdong Provincial Key Laboratory of Advanced Biomaterials (2022B1212010003).

## Author contributions

C. Mi and X.Z. contributed equally to this work. D.J. and C. Mi conceived the project. D. J., C. Liu, C. Liang, and C. Mi supervised the research. X.Z, C.Y., C. Mi, J.W., X.C and Y.L. prepared the animal model and performed the imaging. S.W., Z.Y., C. Mi, and Z.G. carried out the 3D imaging. C. Mi, C. Ma, P.Q., and W.W. synthesized and characterized the nanocrystals. C. Mi, D.J., C. Liu, and C. Liang conducted the research experiments and data analysis. C. Mi and D.J. prepared the figures, and supplementary materials and wrote the manuscript with input from other authors. J.L., J.Z. and M.G. helped with the revision of the paper. All authors participated in the discussion of the results.

## Competing interests

The authors declare no competing interests.
