## [Peer Review File · Nature Communications]

Bone Disease Imaging through the Near-Infrared-II WindowEditorial Note: Parts of this Peer Review File have been redacted as indicated to remove third-party material where no permission to publish could be obtained.

REVIEWER COMMENTS

Reviewer #1 (Remarks to the Author):

1. THE AUTHOR DID NOT USED PRIOR ART ON 4 OPTICAL WINDOWS BY Pu and Shi; and work on bone by Soridillos

2. Moreover they only used window 2 not the best window #3 for imaging bone.

They use thin samples , they could used window 1 and deep vible to get the same result.

The papers does not warrty publication until the aruthoe compare window 3 with 2 and window 1 with 1700 nm and 800 nm to show that their work some difference

Reviewer #2 (Remarks to the Author):

In this work, the authors developed an intravital 3D and high-resolution planar NIR-II (1000-1700 nm) imaging instrumentation for bone disease diagnosis. The authors revealed that polyethylene glycol coated lanthanide-doped nanocrystals (LnNCs) can be passively transported by endothelial cells and macrophages from the blood vessels into bone marrow. Thus, NIR-II imaging of several skeleton diseases (e.g., bone defects, rheumatoid arthritis and osteoarthritis) can be achieved based on these LnNCs. These results sound interesting to the researchers working in this field. However, some critical points in this manuscript should be further clarified before publication in Nature Communications.

1. The authors mentioned that “We therefore developed an intravital 3D and high-resolution planar imaging instrumentation for bone disease diagnosis” in the abstract. However, the 3D imaging is only displayed for NIR-II imaging of normal mice bones. I suggest the authors to provide 3D imaging of bone disease to better show the advantages of the proposed imaging instrumentation.

2. The authors claimed that a series of skeleton diseases can be accurately diagnosed through the NIR-II imaging based on the optical signal of LnNCs. However, these LnNCs were passively transported by endothelial cells and macrophages from the blood vessels. Thus, complex biological interferences and inhomogeneous probe distribution may result in detection deviations and false results. How to surmount such a challenge in skeleton disease diagnosis?

Reviewer #3 (Remarks to the Author):

Chao et al. demonstrate an imaging platform for visualization of bone tissue using light excitation and emission at the NIR-II window (1000-1700 nm), which is expected to penetrate deeper in tissues. To image bone, they further synthesize a nanocrystal probe NaYbF₄: Er³⁺, Ce³⁺ @NaYF₄ (ErNCs). Delivery of the probe via circulation leads to accumulation of the nanocrystals in bone marrow, thereby effectively imaging bone tissue for about a month. Nanocrystals are further shown to be mostly engulfed by macrophages in bone marrow and the infra-red signal disappears after flushing out the bone marrow. To validate the imaging approach for intravital imaging, various animal models are used to show the reconstructed bone, bone defects and the joints in normal and diseased state. Overall, the study presents a new indirect approach for in vivo imaging of bone tissue at a relatively low resolution. Some specific comments for improvement of the presentation of the manuscript.

1. Imaging bone tissue using infra-red nanocrystal can be better rationalized. More information is needed to explain the bone marrow targeting effects of the nanocrystals.
2. Based on images shown in the manuscript, the nanocrystals are localized in macrophages and bone marrow cells. It is not clear to me why they only illuminate bone tissue but not other tissues or organs where macrophages are prevalent?
4. The specificity of these nanocrystals can be better explained.
5. My understanding is that the bone images obtained by this method are not the result of the direct measurement of bone tissue but rather an indirect backlighting effect of the bone marrow cells that phagocytize the nanocrystals. This needs to be better explained.
4. Is it possible to image the dynamics of macrophages in vivo using this method? How this approach can be improved to image bone cells or macrophages at a higher resolution and in

greater details in vivo.

5. The advantages of this method over the traditional MicroCT method for visualization of bone tissue need to be better explained.

6. The images in each figure could be better organized for easy understanding.

Point-by-point response letter to the reviewers:

We would like to thank three reviewers and for taking their time and writing constructive comments to improve the quality of our work. Following the advice, we have conducted new experiments and analyses, and completed the revision with a point-by-point response to the three reviewers.

Our detailed responses (in blue) to the reviewers' comments (in black) are shown below, and the main text change in the paper (in red) below.

Reviewer #1 (Remarks to the Author):

1. THE AUTHOR DID NOT USED PRIOR ART ON 4 OPTICAL WINDOWS BY Pu and Shi; and work on bone by Soridllos

Response: Thanks for raising this good question.

We have carefully referred to the previous works by the mentioned authors (Journal of Biomedical Optics 2014, 19, 056004; Journal of Biomedical Optics 2015, 20, 030501; Journal of Biophotonics 2016, 9, No. 1-2, 38-43; Journal of Biomedical Optics 2017, 22, 045002; Proc. of SPIE 2015, doi: 10.1117/12.2051917), and we noted that two key points may cause the misunderstanding from the reviewer on our paper.

First, the works by Pu, Shi and Sordillo used the NIR light source to light up the tissues directly for ex vivo imaging, with no biofluorescent probe or contrast agent used. They provided imaging results on the bare bones without overlying tissue or covered by chicken tissue slice with 1-2 mm thickness (Proc. of SPIE 2015, doi: 10.1117/12.2051917).

In our work, we proposed the in vivo NIR-II bone imaging by the designed bone-targeting NIR-II probe, and we have demonstrated the feasibility of this method for in vivo bone disease diagnoses, including cortical bone defect, osteophyte, hyperostosis, osteoarthritis, synovitis, and rheumatoid arthritis.

Second, the mentioned works defined optical windows differently. They defined the first NIR optical window as 650-950 nm, the second as 1100-1350 nm, the third as 1600-1870 nm, and the fourth as 2100-2350 nm and centered at 2200 nm, while the generally accepted definition on NIR-II optical window for bioluminescence imaging is from 1000 nm to 1700 nm (Nature Materials 2016, 15, 235-242; Nature Nanotechnology 2021, 16, 1011-1018; Nature Biomedical Engineering 2022, 6, 754-770). Thus, we did not follow the mentioned works by the reviewer.

2. Moreover they only used window 2 not the best window #3 for imaging bone. They use thin samples, they could use window 1 and deep viable to get the same result.

The papers does not warrty publication until the aruthoe compare window 3 with 2 and window 1 with 1700 nm and 800 nm to show that their work some difference

Response: We thank this reviewer for this question.

Figure R1. The quantum efficiency response of the InGaAs camera (Photonic Science) used in our work.

In fact, we used two optical channels for in vivo bone imaging in our work (please see Figure 1b), including the ~1550 nm channel from fluorescence probe ErNCs and the 1060 nm from the main emission of NdNCs. We chose the ~1550 nm optical window because of the much higher imaging contrast and resolution, the strong ~1550 nm emission of ErNCs, and the high quantum efficiency of the used camera around ~1550 nm (please see Figure R1).

We respectfully disagree with this reviewer on the point of “they use thin samples, they could used window 1 and deep vible to get the same result”. On one hand, we did not use thin samples in our work, we used living mice for in vivo bone imaging, and the skeletons inside the body at different depths can be all clearly resolved (please see Figure 1a). On the other hand, the comparison in Figure 1b proves it is not possible to use “window 1 and deep vible to get the same result”, as the shorter wavelength in these two optical windows will lead to stronger light scattering and much lower imaging resolution and contrast compared to ~1550 nm optical imaging.

We also carried out a standard ex vivo test to compare the imaging resolution at different wavelength, as shown in Figure R2, the results quantitatively displayed the imaging resolution by a full width at half maximum (FWHM) as 1.05, 0.86, 0.78 and 0.64 mm for the 1000, 1060, 1340, and 1550 nm optical channel, respectively,

Figure R2. Comparison on the imaging resolution by different optical channels. (a) The NIR-II fluorescence images of glass capillary pairs covered by the 2 mm thick 1% fat emulsion. The glass capillary pairs are filled by the clinically approved dye ICG (~1020 nm emission), NdNCs (~1064 nm emission and ~1340 nm emission respectively) and ErNCs (~1550 nm emission), respectively. Scale bar: 5 mm. (b) Normalized signal intensity profiles and the full width at high maximum (FWHM) across the cross-sections indicated by dash lines in (a), respectively.

Moreover, the imaging at 1700 nm as the reviewer mentioned cannot compare with the ~1550 nm imaging in our work, because the used camera has a much lower quantum efficiency around 30% at 1700 nm than the quantum efficiency around 79% at 1550 nm (please see Figure R1).

We have added the Figure R2 as the new Figure S2 in the revised supplementary information.

Reviewer #2 (Remarks to the Author):

In this work, the authors developed an intravital 3D and high-resolution planar NIR-II (1000-1700 nm) imaging instrumentation for bone disease diagnosis. The authors revealed that polyethylene glycol coated lanthanide-doped nanocrystals (LnNCs) can be passively transported by endothelial cells and macrophages from the blood vessels into bone marrow. Thus, NIR-II imaging of several skeleton diseases (e.g., bone defects, rheumatoid arthritis and osteoarthritis) can be achieved based on these LnNCs. These results sound interesting to the researchers working in this field. However, some critical points in this manuscript should be further clarified before publication in Nature Communications.

1. The authors mentioned that “We therefore developed an intravital 3D and high-resolution planar imaging instrumentation for bone disease diagnosis” in the abstract. However, the 3D imaging is only displayed for NIR-II imaging of normal mice bones. I suggest the authors to provide 3D imaging of bone disease to better show the advantages of the proposed imaging instrumentation.

Response: We appreciate this reviewer’s constructive suggestions.

We have shown the in vivo 3D imaging on normal bones, but not on bone disease diagnosis for three reasons.

First, the 3D imaging is not suitable for the designed time series experiments on long-term disease monitoring in this work, as the 3D bone imaging is not in real time, and one 3D image costs over one hour for raw data acquisition and reconstruction. Moreover, the mice should remain completely still to guarantee the consistency of all 61 pictures for one 3D image reconstruction, so that deep anesthesia is necessary during the whole test. However, the long anesthesia time will increase the mortality rate of the mice a lot.

Second, the planar imaging used for diagnosis can achieve higher resolution than the 3D imaging, as the locations of some pixels may be distorted during the 3D reconstruction by the current algorithm.

Third, the planar image can provide more rapid and concise results for disease diagnosis, which is clinically preferred. The results in Figures 3-5 also prove the two-dimensional image is competent for the bone disease diagnosis. Although the 3D image can more realistically show the whole bone morphology than planar image, but no more key information as it is reconstructed from the planar images.

[figure redacted]

Figure R3. (a) The 3D reconstruction image of mice skin and vessel networks. (b) Vessels with colors that represent their depth under the skin. (Opto-Electron Adv 2023, 6, 220105)

More recently, we just published a new method for 3D NIR-II animal imaging (Opto-Electron Adv 2023, 6, 220105), which can realise not only an axial resolution of 220

μm but also the depth resolution within more than 1 mm (please see Figure R3). We believe this new method could give some valuable reference on this question.

To avoid misleading, we have changed this sentence “we therefore developed an intravital 3D and high-resolution planar imaging instrumentation for bone disease diagnosis” to “we therefore developed the high-resolution NIR-II imaging method for bone disease diagnosis, including the 3D bone imaging instrumentation to show the intravital bone morphology” in Page 2 of the revised manuscript.

2. The authors claimed that a series of skeleton diseases can be accurately diagnosed through the NIR-II imaging based on the optical signal of LnNCs. However, these LnNCs were passively transported by endothelial cells and macrophages from the blood vessels. Thus, complex biological interferents and inhomogenous probe distribution may result in detection deviations and false results. How to surmount such a challenge in skeleton disease diagnosis?

Response: We thank this reviewer for this good question.

First, the light scattering in the biological tissue will cause the uniform distribution of signal intensity, please see Figure R4 in the next page. Although the 1550 nm light at NIR-II range could surpass the scattering rather than short wavelength emissions, the 1550 nm light from body will still be averaged after the in vivo transmission.

Second, bone marrow, together with liver, spleen, and lymph nodes, contain high numbers of macrophage, as they are the macrophage progenitor production and amplification sites (Nature Materials 2014, 13, 125-138). The transport of LnNCs by macrophage will be very frequent and efficient in bone marrow, so that the wide distribution of LnNCs in marrows could guarantee the consistency in imaging on different parts of bone.

Figure R4. The NIR-II imaging on the mouse tibia before (left, in vivo) and after dissection (right, bare) by the 1550 nm emissive ErNCs. The bare tibia contains more imaging details than the tibia in vivo, as indicated by the arrows.

Third, the cortical bone has much lower optical transmittance than the soft tissue. As a result, when the lesion at bone leads to the bone structure or thickness change, the NIR-II signal intensity will change with it obviously as a diagnosis proof. This is proved by Figure 5, it shows the bone structure change caused by osteophyte in the millimeter scale can be observed clearly, and the NIR-II signal has dropped by more than half because the hyperostosis increased the bone thickness by $\sim 100\ \mu\text{m}$. All these results reach an agreement with the microCT examination, which proves the accuracy of this method.

Reviewer #3 (Remarks to the Author):

Chao et al. demonstrate an imaging platform for visualization of bone tissue using light excitation and emission at the NIR-II window (1000-1700 nm), which is expected to penetrate deeper in tissues. To image bone, they further synthesize a nanocrystal probe NaYbF₄: Er³⁺, Ce³⁺ @NaYF₄ (ErNCs). Delivery of the probe via circulation leads to accumulation of the nanocrystals in bone marrow, thereby effectively imaging bone tissue for about a month. Nanocrystals are further shown to be mostly engulfed by macrophages in bone marrow and the infra-red signal disappears after flushing out the bone marrow. To validate the imaging approach for intravital imaging, various animal models are used to show the reconstructed bone, bone defects and the joints in normal and diseased state. Overall, the study presents a new indirect approach for in vivo imaging of bone tissue at a relatively low resolution. Some specific comments for improvement of the presentation of the manuscript.

1. Imaging bone tissue using infra-red nanocrystal can be better rationalized. More information is needed to explain the bone marrow targeting effects of the nanocrystals.

Response: We appreciate this reviewer's constructive suggestions.

We use infra-red nanocrystals ErNCs for bone imaging is because the deep tissue penetration of its ~1550 nm emissions and the passive bone-targeting property, which allow us to achieve the in vivo bone imaging and disease diagnosis.

Figure R5. In vivo NIR-II imaging on mice blood vessel networks after 10 minutes (min) post injection (p. i.) with ErNCs of ~1550 nm emission (left), and NdNCs of ~1060 nm main emission (right), respectively.

Due to the strong light scattering and absorption in biology tissue are wavelength dependent, the visible or short infra-red wavelength light can only realise quite lower image contrast and resolution than the 1550 nm emissions of ErNCs (please see the comparison in Figure 1b for bone imaging, and Figure R5 for blood vessels imaging).

The bone marrow targeting effects of the nanocrystals is a result of the active uptake and transport by marrow cells, thus it is a passive targeting process.

On one hand, even after the nanocrystals were modified by PEG on surface as a camouflage to escape from the hepatic clearance, the plasma proteins (e.g., immunoglobulins, adhesion mediators, complement proteins) can act as opsonins to bind with the nanomaterials, leading to efficient clearance by the mononuclear phagocyte system (MPS). Thus, the nanocrystals will still be captured mainly by the liver and spleen because of the MPS in these organs (see Figure R5).

On the other hand, the MPS also exists in bone marrows, and consists of macrophage, committed stem cell, monoblast, and so on, which could also capture and transport the nanocrystals. More importantly, the slow blood flow and the dense network of blood capillaries in the bone marrows, and the cortical bone as a shell to form an enclosure space, together makes the nanomaterials difficult to escape. As a result, the nanocrystals that transferred to the bone marrow by blood circulation will be endocytosed by local macrophages. In this way, the passive bone-targeting of the nanocrystals can be achieved.

In Page 7 of the revised manuscript, we explained this passive bone-targeting property as: This is because plasma proteins (e.g., immunoglobulins, adhesion mediators, complement proteins) may act as opsonins to bind with the nanomaterials, leading to efficient clearance by the mononuclear phagocyte system in liver, spleen, bone marrows and other organs. Considering the slow blood flow in the dense capillary network inside marrows, the local macrophages take their time to capture the escaped nanoparticles from liver and spleen, and then transport them to a distance in the marrows.

We have added the Figure R5 as the new Figure S3 in the revised supplementary information.

2. Based on images shown in the manuscript, the nanocrystals are localized in macrophages and bone marrow cells. It is not clear to me why they only illuminates bone tissue but not other tissues or organs where macrophages are prevalent?

Response: Thanks for raising this good question.

Figure R6. The in vivo NIR-II imaging on the mouse in supine position after 10 minutes (min), 24

hours (h) and 66 days (d) post injection (p. i.) of ErNCs.

The nanocrystals are also localized in liver, spleen and lung after the tail vein injection, as shown in Figure R6. Because the nanocrystals are typically captured from blood circulation by the mononuclear phagocyte system, including the phagocytosis of macrophages, especially in the liver, spleen and bone marrow.

3. The specificity of these nanocrystals can be better explained.

Response: We thank this reviewer for raising the question on the specificity of the designed nanocrystals ErNCs.

First, the nanocrystals ErNCs can generate an intense emission band around 1550 nm under the excitation of 980 nm laser, such long wavelength can reduce the light scattering in biological tissue. As a result, ErNCs can realise much higher imaging resolution than other fluorescence nano-probes (as discussed in Question 1).

Figure R2. Comparison on the imaging resolution by different optical channels. (a) The NIR-II fluorescence images of glass capillary pairs covered by the 2 mm thick 1% fat emulsion. The glass capillary pairs are filled by the clinically approved dye ICG (~1020 nm emission), NdNCs (~1064 nm emission and ~1340 nm emission respectively) and ErNCs (~1550 nm emission), respectively. Scale bar: 5 mm. (b) Normalized signal intensity profiles and the full width at high maximum (FWHM) across the cross-sections indicated by dash lines in (a), respectively.

We completed a standard ex vivo test to prove this advantage more visually, as shown in Figure R2, the results quantitatively displayed the imaging resolution by a full width at half maximum (FWHM) as 1.05, 0.86, 0.78 and 0.64 mm for the 1000, 1060, 1340, and 1550 nm optical channel, respectively,

Second, the synthesised ErNCs by the coprecipitation method have a solid crystal lattice structure and stable optical properties, which allows the long-term in vivo NIR-II imaging for over 2 months in this work (please see Figure S7).

Third, the ErNCs with PEG capped on surface could ensure the good biocompatibility

for in vivo imaging (please see Figures S8 and S10), and can be excreted gradually after the injection (please see Figure S9).

4. My understanding is that the bone images obtained by this method are not the result of the direct measurement of bone tissue but rather a indirect backlighting effect of the bone marrow cells that phagocytize the nanocrystals. This needs to be better explained.

Response: We thank this reviewer for raising this question.

We agree with this reviewer on this explanation of the NIR-II bone imaging. Indeed, the NIR-II bone imaging depends on the backlighting effect from the bone marrows, this is an optical imaging method by using fluorescence nano-probe to label the target. This is different from the X-ray or CT bone imaging, in which the X-rays penetrate the body and then the remain X-ray intensity pattern after the absorption and scattering by different organs can resolve the internal structure directly.

5. Is it possible to image the dynamics of macrophages in vivo using this method? How this approach can be improved to image bone cells or macrophages at a higher resolution and in greater details in vivo.

Response: Thanks for this good question.

It is quite challenge but possible to image macrophages or other cells in marrow by optimising the proposed in vivo NIR-II imaging method, as the current method in this work could achieve the fluorescence labelling of macrophages after the endocytosis of ErNCs.

First, to image the cells at higher resolution, we need to build up an optical microscope system with high amplification times, as the current optical test system is designed for animal body imaging, with no image amplification function required by micro- or nano-scale imaging. Moreover, the imaging resolution could be improved by using objective lens with high numerical aperture for the microscope.

Second, it must be considered that the NIR-II signal from individual cells is much weaker than the signal from the whole bone. Thus, to optimise the current bone imaging method for marrow cell in vivo imaging, we also need to pretreat the cortical bone (such as removing a small piece of cortical bone and build an inspection window) to enhance the outcoming NIR-II signal, or the fluorescence from the marrow cells could be too weak for microscopy imaging.

[figure redacted]

Figure R7. In vivo dynamic imaging on the process of macrophage migration and nanoparticle uptake by macrophage. (ACS Nano 2022, 16, 4, 6080-6092)

Furthermore, we here refer to one reference cited in our work to show the dynamic imaging on how the macrophages transport nanoparticles in a tumor (ACS Nano 2022, 16, 4, 6080-6092), as shown in Figure R7. With a fluorescence microscopy method at visible optical window, this paper studied in detail how the macrophages capture and

transport the nanoparticles inside a tumor. We believe this study could give some reference on this question.

In the part of the conclusion and outlook in the revised manuscript, we looked forward to the cell kinetics of nanoparticles uptake in bone marrow as: **The rapid progress made in high-resolution optical imaging systems and high efficiency optical materials, as well as the ongoing efforts in studying in vivo specific targeting of nanoparticles with different sizes and surface conditions with improved body clearance, including the cell kinetics during the capture and transport of nanoparticles in marrow, will continue to advance the field of NIR-II imaging towards pre-clinical and clinical translations.**

6. The advantages of this method over the traditional MicroCT method for visualization of bone tissue need to be better explained.

Response: We thank this reviewer for this suggestion.

Compared with MicroCT, the proposed NIR-II method has the advantages of much shorter test time, real-time imaging, no ionizing radiation and more flexible use.

First, the NIR-II imaging usually uses an exposure time of hundreds of milliseconds to complete one test. However, the MicroCT is much slower, as it costs 150 minutes for each hind paw test in Figure 4b, and 55 minutes to complete one knee joint test in Figure 5c, and extra time for 3D image reconstruction is also necessary, which is not real-time imaging.

Second, the NIR-II imaging use the 980 nm laser as the light source, with a low excitation power density of 38 mWcm^{-2} , which causes no biohazard. In contrast, the X-ray used in MicroCT to illuminate the body is classified as a “known human carcinogen”, because the strong ionizing radiation could cause DNA damage and leukocyte death.

Third, the proposed NIR-II method could be used in high frequency, as the NIR-II signal by single injection of ErNCs can last for months (please see Figures R6 and S7), and the time series monitoring on bone disease in Figures 3 and 4 have proved the flexibility of this method. However, the microCT or CT cannot be frequently used due to the potential biohazard caused by ionizing radiation (generally no more than 3 times CT scan for one person per year clinically), and it is limited particularly when pregnancy.

In Page 14 of the revised manuscript, we explained the advantages of the NIR-II bone imaging as: **Compared with the conventional X-ray and μ CT techniques, NIR-II bone imaging can not only satisfy the rapid test and frequent usage, but also provide detailed incidence features and achieve the matchable diagnosis results. More prospectively, it is free of radiation risk for regular use and long-term monitoring of the potential bone disorders and treatment progression.**

7. The images in each figure could be better organized for easy understanding.

Response: We thank this reviewer for this suggestion.

We have carefully rearranged each figure in the manuscript for easy understanding,

please refer to the revised manuscript.

REVIEWER COMMENTS

Reviewer #2 (Remarks to the Author):

The article, in its present form, is much improved with respect to its previous edition. All the questions I raised have been well addressed or explained. I recommend it for publication without further revision.

Reviewer #4 (Remarks to the Author):

This Nature communications submission by Chao Mi et al. presents the development of a new NIR-II emitting fluorescent probe and establishes a high-resolution NIR-II imaging method for visualizing bone tissue. Firstly, while the authors claim to have successfully conducted multi-view images of bone tissue, these images appear to be rough and only provide a faint representation of the bone. It is crucial for the authors to investigate whether their method is actually capable of visualizing the microstructure of bone tissue, including cortical bone, trabecular bone, and collagen orientation. This could be achieved by conducting Z stack imaging using a multi-photon microscope. Additionally, Figure 2c does not convincingly demonstrate the uptake of the NIR-II probe by the macrophages. It is essential for the authors to examine the colocalization between macrophages and the probe by performing 3D imaging at a large scale using tissue clearing methods. This would provide stronger evidence for the probe interaction with macrophages. Secondly, as mentioned by a previous reviewer, the authors should provide 3D imaging of bone diseases. Numerous methods for completely capturing the field of view of anesthetized live mice in multiphoton bone imaging have already been established. The authors should improve their observation procedure and determine whether their method is applicable for visualizing the onset and progression of bone diseases over a longer time course. Lastly, in this study, the authors prepared bone tissue by incising the skin and expose bone tissue. To strengthen the advantages of this method over traditional microCT imaging, it is crucial for the authors to observe intact bone tissue in a non-invasive manner. This would further highlight the potential of their approach. Overall, this is technically immature, and significant revisions and additional investigation are necessary. At its current state, it does not meet the level expected for publication in Nature Communications.

Point-by-point response letter to the reviewers:

We would like to thank all the reviewers for taking their time and writing constructive comments to improve the quality of our work. Following the advice, we have conducted new experiments and analyses, and completed the revision with a point-by-point response to the three reviewers.

Our detailed responses (in blue) to the reviewers' comments (in black) are shown below, and the main text change in the paper (in red) below.

REVIEWER COMMENTS

Reviewer #2 (Remarks to the Author):

The article, in its present form, is much improved with respect to its previous edition. All the questions I raised have been well addressed or explained. I recommend it for publication without further revision.

Response: We thank this reviewer again for the comments on this work.

Reviewer #4 (Remarks to the Author):

This Nature communications submission by Chao Mi et al. presents the development of a new NIR-II emitting fluorescent probe and establishes a high-resolution NIR-II imaging method for visualizing bone tissue. Firstly, while the authors claim to have successfully conducted multi-view images of bone tissue, these images appear to be rough and only provide a faint representation of the bone. It is crucial for the authors to investigate whether their method is actually capable of visualizing the microstructure of bone tissue, including cortical bone, trabecular bone, and collagen orientation. This could be achieved by conducting Z stack imaging using a multi-photon microscope. Additionally, Figure 2c does not convincingly demonstrate the uptake of the NIR-II probe by the macrophages. It is essential for the authors to examine the colocalization between macrophages and the probe by performing 3D imaging at a large scale using tissue clearing methods. This would provide stronger evidence for the probe interaction with macrophages. Secondly, as mentioned by a previous reviewer, the authors should provide 3D imaging of bone diseases. Numerous methods for completely capturing the field of view of anesthetized live mice in multiphoton bone imaging have already been established. The authors should improve their observation procedure and determine whether their method is applicable for visualizing the onset and progression of bone diseases over a longer time course. Lastly, in this study, the authors prepared bone tissue by incising the skin and expose bone tissue. To strengthen the advantages of this method over traditional microCT imaging, it is crucial for the authors to observe intact bone

tissue in a non-invasive manner. This would further highlight the potential of their approach. Overall, this is technically immature, and significant revisions and additional investigation are necessary. At its current state, it does not meet the level expected for publication in Nature Communications.

Response: We thank this reviewer for raising the good comments.

First, we need to emphasize that, except part of Figure 1e was ex vivo imaging result as contrast, all the other NIR-II bone imaging results shown in the manuscript were captured in living mice, by an in vivo and non-invasive way. We did not incise or remove the skin to expose bone tissue for exposed bone imaging.

The intact murine bones without any tissue processing, inside the living body, including skull, spin, sternum, tibia, phalanx, subchondral bone, rib, and femur, all can be clearly shown by the proposed NIR-II in vivo imaging (please see Figure 1a). Moreover, the bone disease diagnosis, including 1 mm bone defects detection, rheumatoid arthritis, synovitis, osteoarthritis, osteophyte and hyperostosis, were all carried out in living mice non-invasively with our method (Figures 3-5).

Thus, we respectfully disagree with this reviewer on the point of “in this study, the authors prepared bone tissue by incising the skin and expose bone tissue. To strengthen the advantages of this method over traditional microCT imaging, it is crucial for the authors to observe intact bone tissue in a non-invasive manner”. In fact, all the bone NIR-II imaging were based on living animal experiments in our work through an in vivo and non-invasive approach, under same conditions for microCT and X-ray imaging.

Second, on one hand, the proposed NIR-II imaging method have the advantages of deep tissue penetration and high resolution on living body imaging, but it is not a microscopic imaging method in this work. The aim of this method is to realise living body bone disease diagnosis. The mentioned multi-photon microscope is usually for in vitro tissue section pathological imaging (Scientific Reports 2017, 7, 3419). Although multi-photon microscope can be used for living mouse imaging, the skin removal or skull optical clearing is necessary due to the light blocking (Light Sci Appl 2018, 7, 17153; Cell Rep 2017, 18, 1804-1816; Cytometry Part A 2020, 97A, 496-503), and it is limited to observe parietal bones of mice because of the thin thickness, but not other thicker bones (Front Immunol 2019, 10, 596), moreover, the excitation laser power density is much more higher than the NIR-II imaging. On the other hand, the NIR-II probe ErNCs in this work cannot target to the cortical bone, trabecular bone, or collagen (please see Figure 2e), it can target to the marrow cells as explained in the manuscript.

Thus, this method cannot visualize the mentioned microstructure of the cortical bone, trabecular bone, or collagen, but is suitable for a series of bone disease diagnosis with imaging resolution up to 1 mm (Figures 3-5).

Figure R1. 3D confocal microscopic imaging on the stained tibia sections collected from a mouse 36 h after ErNCs@Cy3 injection, including nine zoom-in corresponding XY and XZ sections of interest. Green channel: F4/80 labeled macrophages. Red channel: ErNCs@Cy3. Blue channel: DAPI labeled cell nucleus. Scale bar: 80 μm .

Third, according to the constructive suggestion on macrophages targeting by the reviewer, we carried out the 3D confocal microscopy imaging to confirm the colocalization between macrophages and the ErNCs probe (Z-stacks of 40 μm in thickness with step of 2 μm were taken, and x-y resolution is 0.624 μm). Please see Figure R1, all the zoom-in XY and XZ section images from the large area show the signal overlapping and prove the uptake of ErNCs by macrophages.

[figure redacted]

Figure R2. In vivo dynamic imaging on the nanoparticle uptake by macrophage. (ACS Nano 2022, 16, 4, 6080-6092)

Furthermore, we here refer to one reference cited in our work to show the dynamic imaging on how the macrophages transport nanoparticles in a living mouse (ACS Nano 2022, 16, 4, 6080-6092), as shown in Figure R2. This paper studied in detail how the macrophages capture and transport the nanoparticles in vivo.

We have added Figure R1 as the new Figure 2c in the revised manuscript, and the previous Figure 2c has been used as the new Figure S8 in the supplementary information. The corresponding text has been revised as “by co-localization 3D confocal microscopic imaging on stained marrow sections, we confirmed that the Cy3 fluorescence of ErNCs@Cy3 overlaps with the channel of macrophages (Figures 2c and S8) with a Pearson correlation coefficient of up to 0.57, which suggests the macrophage’s uptake of ErNCs” in the revised manuscript.

Fourth, we have not shown the 3D NIR-II imaging on bone disease diagnosis for the following three reasons, as a same question by Reviewer 2.

First, the 3D NIR-II imaging is not suitable for the designed time series experiments on long-term disease monitoring in this work, as the 3D bone imaging is not in real time,

and one 3D image costs over one hour for raw data acquisition and reconstruction. Moreover, the mice should remain completely still to guarantee the consistency of all 61 pictures for one 3D image reconstruction, so that deep anesthesia is necessary during the whole test. However, the long anesthesia time will increase the mortality rate of the mice a lot.

Second, the planar imaging used for diagnosis can achieve higher resolution than the 3D imaging, as the locations of some pixels may be distorted during the 3D reconstruction by the current algorithm.

Third, the planar image can provide more rapid and concise results for disease diagnosis, which is clinically preferred. The results in Figures 3-5 also prove the two-dimensional image is competent for the bone disease diagnosis. Although the 3D image can more realistically show the whole bone morphology than planar image, but no more key information as it is reconstructed from the planar images.

[figure redacted]

Figure R3. (a) The 3D reconstruction image of mice skin and vessel networks. (b) Vessels with colors that represent their depth under the skin. (Opto-Electron Adv 2023, 6, 220105)

More recently, we just published a new method for 3D NIR-II animal imaging (Opto-Electron Adv 2023, 6, 220105), which can realise not only an axial resolution of 220 μm but also the depth resolution of more than 1 mm (please see Figure R3). We believe this new method could give some valuable reference on this question.

In addition, as discussed above, the mentioned multiphoton bone imaging for intravital imaging usually needs tissue preprocessing, and is not suitable for all the bones imaging in different body position, while our method can be used for all the bones imaging and more flexible diagnosis, the application scenarios of the methods are quite different.

To avoid misleading, we have changed this sentence “we therefore developed an intravital 3D and high-resolution planar imaging instrumentation for bone disease diagnosis” to “we therefore developed the high-resolution NIR-II imaging method for bone disease diagnosis, including the 3D bone imaging instrumentation to show the intravital bone morphology” in the revised manuscript.

REVIEWERS' COMMENTS

Reviewer #4 (Remarks to the Author):

The authors have sufficiently addressed the reviewers' concerns.